# Vitamin B12 (Cobalamin): Its Fate from Ingestion to Metabolism with Particular Emphasis on Diagnostic Approaches of Acquired Neonatal/Infantile Deficiency Detected by Newborn Screening

**DOI:** 10.3390/metabo12111104

**Published:** 2022-11-12

**Authors:** Magdolna Kósa, Zsolt Galla, István Lénárt, Ákos Baráth, Nóra Grecsó, Gábor Rácz, Csaba Bereczki, Péter Monostori

**Affiliations:** Department of Pediatrics, University of Szeged, Korányi fasor 14-15, 6720 Szeged, Hungary

**Keywords:** vitamin B12 deficiency, cobalamin deficiency, newborn screening, diagnostic approaches

## Abstract

Acquired vitamin B12 (vB12) deficiency (vB12D) of newborns is relatively frequent as compared with the incidence of inherited diseases included in newborn screening (NBS) of different countries across the globe. Infants may present signs of vB12D before 6 months of age with anemia and/or neurologic symptoms when not diagnosed in asymptomatic state. The possibility of identifying vitamin deficient mothers after their pregnancy during the breastfeeding period could be an additional benefit of the newborn screening. Vitamin supplementation is widely available and easy to administer. However, in many laboratories, vB12D is not included in the national screening program. Optimized screening requires either second-tier testing or analysis of new urine and blood samples combined with multiple clinical and laboratory follow ups. Our scope was to review the physiologic fate of vB12 and the pathobiochemical consequences of vB12D in the human body. Particular emphasis was put on the latest approaches for diagnosis and treatment of vB12D in NBS.

## 1. Introduction

VB12, also called cobalamin (Cbl), is present in several forms in nature. Cbl deficiency might be defined as absolute deficiency of vB12, or functional deficiency with the metabolite profile characteristic for inadequate cellular function of Cbl dependent enzymes (reviewed in several articles recently [1,2,3,4]). This review focuses on a nowadays still underdiagnosed condition, namely infantile acquired vB12D of maternal origin. Correct diagnosis of this condition is often challenging especially when considering the great inequity in the technical, financial background of different NBS programs and the differences in the socioeconomic status of pregnant women across the globe. By reviewing recent literature (with particular focus on the period 2018–2022), we aimed to discuss the biochemical background of the condition leading to the observed metabolic disturbances, the symptoms, prevalence, available diagnostic methods and guidelines for NBS, as well as options for treatment and follow-up.

## 2. From Uptake to Utilization

### 2.1. Chemistry and Dietary Source

From the chemical point of view, the corrin ring of the Cbl molecule is in covalent bond with the central cobalt ion (of various oxidation state). Above this plain, the beta-axial ligand bound to the cobalt ion determines specific reactions in which Cbl might be involved. Based on this latter ligand, molecules with distinct biochemical functions are distinguished like adenosyl-Cbl (AdoCbl), methyl-Cbl (MeCbl), hydroxy-Cbl (OHCbl), cyano-Cbl (CNCbl), and glutathionyl-Cbl (GSCbl). Most of these molecules are synthesized after intracellular processing of Cbl. AdoCbl and MeCbl are essential cofactors of methylmalonyl Coenzyme A (MMCoA) mutase (MUT) and methionine synthase (MS), respectively, enzymes in the intermediate metabolism of the human body.

Cbl rich diet includes animal products such as meat, fish, dairy products, liver and egg. Strict vegan or vegetarian diet without supplementation might lead to vB12D, already after 4 weeks of dietary change from omnivore to strict vegan diet [5,6]. In a study involving Dutch pregnant women (at 34–36 weeks of pregnancy) of mixed population including omnivore, lacto-ovo vegetarians and vegans, the main source of Cbl were dairy products, while eggs as Cbl source represented the minority of dietary Cbl source [7].

### 2.2. Fate of Cbl in the Gastrointestinal Trackt

The process from ingestion to ileal absorption is demonstrated on Figure 1 and Figure 2.

During absorption (Figure 2), the intrinsic factor (IF)-Cbl complex connects to its receptor cubam which is a heterodimer of the proteins amnionless and cubilin. The IF-Cbl complex is internalized by receptor mediated endocytosis. Later, cubam is recycled to the cell surface, while in parallel the pH of the vesicle turns acidic and becomes a lysosome. Low pH allows the dissociation of IF from Cbl, the released IF is degraded (reviewed by [9]). Intracellular trafficking from the lysosome to the cytoplasm involves the proteins LMBD1 and ABCD4. After its synthesis, ABCD4 protein is directed to the endoplasmic reticulum membrane, where it interacts with LMBD1 protein, this latter enabling the transition of the LMBD1-ABCD4 complex to the lysosomal membrane. ABCD4 is the key transporter of Cbl from the lysosomes to the cytosol where it is further processed in most cells [10]. Both proteins show high expression levels in several tissues, including small intestines [11].

Intracellular processing differences between gut epithelial cells or other peripheral, non-polarized cells are barely discussed in the literature. It is suspected that Cbl either undergoes processing to the biologically active cofactor forms (AdoCbl, MeCbl) that are finally used by enterocytes for specific reactions of metabolism, or passes the cell without major modifications (transcytosis) to enter the portal circulation (reviewed by [9]). The transport toward the blood stream occurs via ABCC1 (alternative name: MRP1 (multidrug resistance protein-1)), an ABC transporter located in the basolateral membrane of enterocytes. MRP1 binds and exports free Cbl. In the extracellular matrix free Cbl is bound to the apo-transcobalamin-2 molecule (TC2, Uniprot search term: P20062 or TCO2_HUMAN) which is excreted by cells to the extracellular matrix. The TC2-Cbl complex is later referred to as holo-transcobalamin (HoloTC) [8,9,12,13]. Experiments on bovine aortic endothelial cell cultures showed that endothelial cells excrete a significant concentration of TC2 and also express the HoloTC receptor (CD320) with which a complex Cbl uptake machinery is offered. The fate of HoloTC in endothelial cells might be transcytosis (HoloTC leaves the cell unmodified, ~5–10% of excreted Cbl), exocytosis asfree Cbl (via ABCC1, ~90–95% of excreted Cbl), metabolic modification into ready-to-use cofactors for enzymatic processes, or HoloTC is left intact within the cells. The possibility of choosing different intracellular fates is equally open to Cbl molecules with different moieties (AdoCbl, MeCbl, CNCbl, OHCbl) [14].

Some fraction of intestinal Cbl might be bound to luminal free TC2 and absorbed via CD320 receptor according to studies on murine small and large intestine, and human polarized colorectal cancer cell lines. Another, rather small fraction of Cbl is not absorbed at all, affecting microbiota metabolism and pathogenicity in large intestines. An even smaller amount is eliminated by the stool [15,16].

### 2.3. Intracellular Fate, Transport in Blood and Storage

Cytosolic processing of Cbl after release from the lysosome is a multistep procedure, in which apoenzymes bind and chaperonate different forms of Cbl (Figure 3). Many questions are yet to be answered, such as interaction between these proteins and with Cbl, their substrate specificity considering the beta-axial ligand of the corrin ring or oxidation state of the cobalt ion, or transport and recycling between intracellular organelles (from cytosol to mitochondria). Each of the proteins involved in intracellular processing is linked to a certain inherited Cbl disorder (Table 1). The aim of the process is to yield the proper amount of active coenzymes required for the reactions of the MUT and MS enzymes.

Transport of Cbl in blood occurs either in form of HolotTC (~20–25% of circulating Cbl) or TCN1-Cbl (~75–80% of circulating Cbl). HoloTC is considered the biologically available (“active”) Cbl in the circulation that can enter non-polarized cells via CD320 mediated endocytosis. On the other hand, TCN1-Cbl is considered to transport Cbl to hepatocytes, and allowing the storage or mobilization from hepatic Cbl buffer depending on Cbl storage levels of the body [4,12].

Kidneys might also serve as a Cbl reserve. The main receptor for reabsorbing Cbl in proximal tubular cells is megalin. Apart from this, cells also exhibit a remarkable intracellular TCN2, and apical CD320 expression in murine experiments. Thus excreted TCN2 can form HoloTC taken up by their receptors at the luminal surface of epithelial cells [16,17].

## 3. MUT and MS Catalyzed Reactions

The MUT enzyme (EC 5.1.99.1) is catalyzing the reaction in which MMCoA is converted to succinyl CoA (SuCoA); while the enzyme MS (EC 2.1.1.13) catalyzes the reaction where homocysteine (HCY) is converted to methionine (Met) (Figure 3). AdoCbl and MeCbl, respectively are the active Cbl cofactors of the latter enzymes. The above reactions yield substrates and regulatory molecules for a variety of reactions such as the anaplerotic molecule succinate for the tricarboxylic acid cycle or the methyl donor S-adenosyl methionine (SAM) for DNA methylation during epigenetic modifications. In case of insufficient availability of AdoCbl and MeCbl, clinical symptoms may be present as a consequence of the disturbed cellular metabolism. The impaired function of both enzymes might occur in Cbl deficiencies—regardless of the inherited or acquired origin of the condition—resulting in hematologic and/or neurologic symptoms of various degree (reviewed by [13]).

## 4. Pathobiochemical Consequences of vB12D

Considering the bone marrow, inadequate availability of MeCbl leads to accumulation of HCY and decreased Met, while the methyl donor methyl-THF is accumulating, thereby trapping the methyl group from other cellular methylation processes. This leads to abnormal nucleotide synthesis resulting in the histopathologic picture of macrocytic anemia (reviewed by [2]). Clinical descriptions from patients of various age show that hematologic effects are not necessarily present in vB12D [2,18,19,20]. According to a recent review on pernicious anemia (PA, a condition which evolves due to antibodies against IF, thereby preventing the intestinal absorption of Cbl), macrocytosis might be present years before development of anemia. On the other hand, ~30% of these patients never develop macrocytosis; concomitant iron deficiency anemia with coprevalence of ~20% might be a significant contributor to this. Interestingly, vB12D might also present in the form of conditions severely altering hemopoesis, such as pseudo-leukemia mimic syndrome or bone-marrow failure-like syndrome [21].

Neurocognitive symptoms might be linked to impaired methylation → disturbed myelination → increased inflammatory cytokine axis, leading in extreme cases to the atrophy of the spinal cord or cerebral tissue in different degree. The spectra of childhood neurologic symptoms associated with Cbl deficiency might include convulsion, muscular hypotonia, developmental delay, intellectual disability, dizziness, tingling sensation, syncope, or blurred vision [19]. Most of the functional neurologic disturbances can be reversed by parenteral vitamin B12 supplementation even in adults [18]. The effect of supplementation is also promising in pediatric children with autism spectrum disorder [22]. Coexistence of both hematologic and neuropsychiatric symptoms is not a prerequisite in vB12D.

## 5. Diagnostic Criteria of Vitamin B12 Deficiency

Standardized international definition or consensus guidelines for a vB12D are lacking. Laboratory evaluation of a vB12D allows determination of at least four distinct laboratory markers: total abundance of vB12, HoloTC, MMA and HCY [2,3,20]. There seems to be no single “perfect” biomarker; rather the combination of these markers in “four-variable”, “three-variable” (e.g., vB12 + HoloTC + MMA, HoloTC + MMA + HCY or vB12 + MMA + HCY) or “two-variable” (most often HoloTC + MMA) approaches, dependent on the laboratory infrastructure, financial sources and patient population [23,24]. Methylcitric acid (MCA) is an additional important biomarker in screening and diagnosis of propionic aciduria (PAuria) and MMAuria but is reportedly less useful in Cbl deficiencies and is rarely determined by general clinical laboratories [25].

### 5.1. VB12 Assays and Result Interpretation

Among analytical methods on vB12 determination, microbiological assays were the first to have found utility in the clinical laboratory, due to acceptable sensitivities and specificities [3]. This time-consuming technique does not provide a direct estimation of the vitamin levels but functional information that is dependent on utilization of the vitamin by the microbe used [26]. Of several strains, Lactobacillus leichmannii (ATCC 7830) was used most frequently due to the robustness of the assay [3,27]. Radioisotope-based dilution techniques were also published [28]. However, automated methods are most widely used currently for the assessment of vB12 status due to their high throughput and less laborious nature [3]. Competitive-binding luminescence-based assays (CBLA) for total (i.e., protein-bound plus free) vB12 are available since the early 1990’s and include electrochemiluminescent (Roche), chemiluminescent (Siemens, Abbott, Beckman Coulter, VITROS), enzyme-linked fluorescent (earlier Siemens and Abbott assays), or colorimetric techniques [3,29]. Serum vB12 assays are calibrated independently by manufacturers with traceability to an internally manufactured standard material rather than an internationally certified reference material [3]. This often results in poor agreement between commercially available assays for serum vB12, despite the ratification of The World Health Organization International Standard for serum B12 (03/178) in 2007 as a consensus of vB12 protein-binding assays [3,30]. Thus, assay-specific cutoffs are still generally used for the different vB12 assays, which hamper comparability of results from different settings, regions or countries [3].

Serum Cbl concentrations alone are weak indicators for functionally relevant intracellular Cbl deficiency [2]. A functional vB12 deficiency can be present in patients with apparently normal or even high serum vB12 levels, for example in the case of high levels of vB12-binding proteins [3]. The presence of macro-transcobalamin can reportedly increase vB12 results in various assays [31]. Erroneously high or normal total vB12 concentration may be caused by interfering anti-IF antibodies in pernicious anemia (where vB12 levels would be expected to decrease). This condition might also affect mothers of vB12 deficient infants detected by NBS [2,3,32,33].

### 5.2. HoloTC Assays and Result Interpretation

The second marker commonly used is HoloTC, the form of vB12 that is taken up by cells to meet metabolic demand, explaining the increasing use of assays on this biomarker [3,20]. HoloTC seems to be a helpful parameter to identify problems of absorption and to differentiate them from nutritional Cbl deficiency [2].

The first method for HoloTC was based on the enzyme-linked immunosorbent assay technique (ELISA) [34]. Currently, most clinical laboratories use ELISAs or (electro)chemiluminescent (ECLIA, CLIA) immunoassays for HoloTC [3]. In contrast with commercial automated total methods for vB12 that use assay-specific calibration with no international reference material, calibrators of commercially available assays for HoloTC are traceable back to common frozen primary reference calibrators which are held by Axis-Shield [3]. However, reference values are still strongly dependent on the assay method used [34]. In addition, some rare variants in the transcobalamin gene can reportedly interfere with the HoloTC determination, so that erroneously low HoloTC levels are measured, despite an absence of clinical deficiency and normal levels of all other laboratory markers [3]. In contrast with vB12, HoloTC results are reported to be relatively unaffected by assay interference from high-titer IF antibody concentrations [3]. However, HoloTC may still be erroneously normal in a small subset of patients with pernicious anaemia [35].

### 5.3. MMA Assays and Result Interpretation

The concentration of MMA in serum reflects the availability and utilization of AdoCbl in mitochondria. MMA in fresh or optimally stored urine or blood is considered the most sensitive marker of intracellular, functionally relevant Cbl depletion (with the limitation that urinary MMA is not reliable in individuals with impaired renal function) [2].

The first techniques for the measurement of MMA applied paper chromatography, thin layer chromatography, spectrophotometry and later ELISA, but were superseded by mass spectrometric methods due to markedly improved sensitivities and specificities. Full or semi-automation of liquid chromatography-tandem mass spectrometry (LC-MS/MS) methods for MMA is also possible [3]. Determination of MMA in NBS from dried blood spots (DBS) will be discussed in a separate section.

An elevated MMA is not specific for vB12 depletion or inborn errors of the Cbl metabolism. Among other disorders, a deficiency of the enzymes MUT (total or partial) or acyl-CoA synthetase family member 3 (ACSF3) can also result in MMA levels increased to various magnitudes [25]. Elevated MMA levels have also been reported in people with severe impairment of renal functionality due to impaired elimination [2,20].

### 5.4. HCY Assays and Result Interpretation

The total concentration of HCY in serum reflects the availability and utilization of MeCbl in the cytosol [1]. Historically, HCY was measured by means of paper chromatography, radio-enzymatic determination or ion-exchange amino acid analyzers. Today, automated enzyme immunoassays and high performance-LC methods (HPLC) with fluorescent or electrochemical detection are used, in addition to mass spectrometry coupled with gas chromatography (GC) or LC [3,36]. From the analytical perspective, the measurement of HCY is relatively challenging. The sulfhydryl group of HCY readily reacts with sulfhydryl groups of proteins, another HCY molecule or cysteine to form the corresponding disulfide compound. Therefore, analytical methods either determine free HCY without any reduction step (as in most immunoassays) or total HCY after reaction with a reducing agent (DTE (1,4-Dithioerythritol), DTT (DL-Dithiothreitol) or TCEP (Tris(2-carboxyethyl)phosphine hydrochloride)) to release HCY bound to proteins and other disulfides [18]. In order to avoid artificial alterations in serum HCY levels due to preanalytical issues, blood should be obtained in the fasting state, transported cooled (the freezing pack should not get in touch directly with the blood tube) and centrifuged as soon as possible, preferably within 30 min [37]. Measurement of HCY from DBS in the frame of NBS will be discussed in a separate section.

In addition to inborn errors of the Cbl metabolism, other congenital disorders can also manifest with elevations in serum HCY, including a cystathionine-β-synthase (CBS) or a methylenetetrahydrofolate reductase (MTHFR) deficiency [3]. Elevated HCY values can also occur as a consequence of folate or vitamin B6 deficiency, as well as impaired renal function, hypothyroidism, malignant tumors and certain medications [2,38]. In the aspect of age HCY seems more to be a marker of impaired Cbl metabolism in infants and toddlers, while it is associated primarily with abnormalities of folate metabolism in elderly. In this latter age group MMA is considered as primary marker of Cbl deficient cellular metabolism [1,39].

As a complement to the determination of HCY, a recent article reported an LC-MS/MS method for the determination of SAM and S-adenosylhomocysteine (SAH) in plasma and urine. The determination of SAM and SAH is complicated by the instability of SAM under neutral and alkaline conditions and the naturally low concentration (nanomolar range) of both SAM and SAH in plasma. The clinical applicability of the assay is yet to be determined [40].

### 5.5. General Considerations

Recommendations on the determination of vB12D include in optimal case measurement of serum vB12 or HoloTC combined with two functional markers, MMA and HCY. It is important to emphasize that HCY may also increase in folate deficiency; therefore, additional measurement of folic acid is needed for differential diagnosis [1]. This latter is also included in the mathematic formula for calculating the so called combined vB12 [23]. Cut-off values largely vary between studies. The low cut-off range of total vB12 in different centers was 118–221 pmol/L, for HoloTC 37–50 pmol/L. High cut-off of serum total HCY was in the interval 8–13.6 µmol/L (8 µmol/L when vB12D was determined based on single total HCY measurement without vB12 determination) and MMA in the range of 271–800 nmol/L [1]. Based on the results of the primary and functional markers low vB12, possible vB12D and probable vB12D might be distinguished. These result in subclinical vB12D, vB12D with possible or significant clinical manifestations, respectively [23]. According to the opinion of the authors, clinical laboratories should primarily define their cut-off values based on the diagnostic method they use, and literature data should only serve as orientation point.

## 6. Vitamin B12 Deficiency among Pregnant Women

In most countries, screening of vB12D status is not compulsory during pregnancy [41,42]. Estimating and comparing the prevalence of this condition between geographic regions might be difficult because of the unstandardized definition of the laboratory assessment of vB12D [43]. However, the reported prevalence is relatively high, around 20% worldwide, especially in low-income areas where access to animal food is strongly limited [41,44]. In concordance with this, a positive correlation between the educational level of the mother and vB12 level was also observed [45]. The subclinical vB12D of the mothers is usually diagnosed only when their breastfed infant is identified during NBS with suspected vB12D. In these cases, metabolite elevations in DBS indicative of MMAuria/PAuria followed by further testing may shed light on the hidden vitamin deficiency of the mother [44,46]. Several gene products are involved in absorption, transport and intracellular trafficking of Cbl (Table 1). Nevertheless, the acquired vB12D of the mother results predominantly in vB12D of the breastfed infant [41,44,47]. Secondary causes of vB12D are summarized in Figure 4.

During pregnancy, concentrations of several biochemical markers are lower than those in non-pregnant women [48]. Hormonal changes resulting in physiologic hemodilution might be a factor contributing to this, in addition to fetal consumption and redistribution of the Cbl stores of the body [42]. Attempts to determine cutoff-values of vB12 in early, mid, and late pregnancy show a general tendency of a gradual decline of vB12 toward delivery, followed by a sharp rise during early lactation [48,49]. In parallel, TC1 declines more sharply, while HoloTC is maintained at a relatively stable level, albeit reports on change in HoloTC level during gestation remain controversial [50,51]. In population studies, these changes are accompanied by elevation in functional vB12D markers MMA and HCY [41,42,48,52]. In contrast, a Canadian study conducted on healthy pregnant women (not representing the entire population) noted the decrease of both MMA and HCY toward terminus [53]. According to these, trimester specific cutoffs were suggested for functional biomarkers [52]. Some studies hypothesized low Cbl status to be a side effect of the widespread prophylactic folate substitution in early pregnancy. There was, however, no proven evidence for a mechanistic link between the two. The corrected folate levels seemed to be more a co-occurrence with a non-corrected Cbl status [54].

## 7. Vitamin B12 Deficiency among Newborn

Cbl repletion of the newborn at the time of delivery is substantially determined by the Cbl status of the mother (involving both circulating and functional markers). There is positive correlation in each trimester between the Cbl status of the mother and newborn [55]. Moreover, this relationship (as early as the 18th gestational week), has a significant impact on the Cbl status of the breastfed infant 6 months post-partum [56]. However, even more enduring changes might occur according to a study in which the mid-pregnancy deficient vB12 levels of the mothers were accompanied with impaired cognitive function of their toddler as far as 24 months age [57]. Beside this, elevated HCY itself during the second and third trimester of pregnancy was inversely correlated to neurocognitive outcome of children at 30 months [58]. It seems that Cbl is accumulating in the fetus up till delivery, since serum vB12 concentration might be 2.5-fold higher in the newborn compared to its abundance in its mother [59]. It is yet to be evaluated whether nitrous oxide, an anesthetic gas commonly used during labor, could lead to transient elevations in the markers of a vB12D in NBS [60]. Even if nitrous oxide can irreversibly inactivate Cbl (when bound to MS—Figure 3.) by oxidizing the cobalt ion, only its regular use as drug was reported to be associated with neurologic symptoms in earlier papers [47,61]. However, recent results from the Norwegian NBS program suggested that nitrous oxide applied during labor might lead to vB12D [62]. After birth, infants on formula feeding receive an optimal supplementation of Cbl, while breastfed infants have to rely on the amount of Cbl the mammary glands excrete into the breastmilk, which show different dynamics in pre-term and full-term delivered infants [44,63]. Breastmilk Cbl peaks 4 weeks post-partum for infants born before the 34th gestational week. This is in contrast to full-term newborns where the peak is already at the time of birth, from which a gradual decline can be observed toward day 28 post-partum [63]. Recently, particular attention has been paid to the diagnosis of vB12D in NBS due to its relatively high prevalence, cheap and widely available treatment, and its capability to not only identify infants but also mothers with subclinical vitamin deficiency.

## 8. NBS Protocols and New Candidate Markers

NBS allows early recognition and treatment of numerous inborn errors of metabolism, thus facilitating the prevention of serious morbidity and mortality [64]. DBS is the standard sample type in NBS worldwide due to the small volume of blood needed, the ease of transportation and stability [65]. Expanded NBS took advantage of the implementation of MS/MS, allowing an increase in the number of screened disorders via simultaneous determination of several amino acids and acylcarnitines [64]. Among disorders of the propionate metabolism, PAuria and MMAuria are already included in NBS in numerous European countries and in all US states [66,67].

However, the characteristic alterations of an elevated propionylcarnitine (C3) and/or decreased Met may additionally be suggestive of a vB12D, that was previously considered an incidental finding and inappropriate for an inclusion in NBS until very recently [25,68,69]. The reason was that none of the first-tier parameters is specific for any of these conditions, which results in an elevation of the rate of false-positive results [25]. Even if false-positive samples cannot fully be avoided in NBS, their number has to be kept low to minimize parental anxiety, as well as costs and workload [64]. Moreover, even if C3 and Met are generally measured together with numerous other amino acids and acylcarnitines in a single assay, these parameters are often only evaluated if the country officially screens for these disorders [25].

Recently, an increasing number of publications suggested that an inclusion of a vB12D into NBS may not only be reasonable but feasible too. According to the authors, the proper selection of diagnostic analytes, cutoffs and strategies can facilitate detection of a vB12D. Thus, an inclusion of this condition into NBS is currently under evaluation in some countries including Germany [68,69]. The use of analyte ratios in the screening test such as the ratios Met/phenylalanine (Met/Phe) and C3/acetylcarnitine (C3/C2) can directly improve false-positive rates.

Diagnostic specificity can be further increased by confirmatory assays utilizing chromatographic separation of diagnostically important analytes (including MMA, MCA, HCY and rarely 3-hydroxy-propionic acid) from their isobaric substances, i.e., from those having the same molecular mass [25,68,69]. In addition to succinate, the only metabolite considered earlier as an interference for MMA, further five potential isobaric interferents have been identified recently [70]. The urinary organic acid assay by means of GC-MS performs well in separating isobaric interferences that could compromise diagnosis and an accurate quantification [65]. In contrast with the initial DBS sample being available for the NBS laboratory, urine has to be requested separately, which is associated with marked delays and significant increase of parental anxiety [25]. Thus, it is strongly recommended that confirmatory assays in NBS use the original DBS sample (second-tier tests) [25,64].

The very limited sample amount in DBS raises an analytical challenge for most diagnostic metabolites [65]. Thus, even if derivatization can be avoided for faster turnover times if sufficiently sensitive mass spectrometers are used [71], most assays for HCY, MMA and MCA in DBS use derivatization with n-butanolic HCl or silylated trifluoroacetamides [72,73,74]. HCY can be measured separately from MMA and MCA [71,75,76] or simultaneously in a single assay [28,29,30]. In contrast with MMA, MCA and HCY, vB12 is not determined from DBS but from serum, due to its very low concentration, high molecular mass and complex structure [3].

Considering the diagnostic value of the first- and second-tier markers, a vB12D can present with diverse alterations [25,69]. In a recent publication, the most sensitive first-tier parameters were a low Met/Phe (positive in 23 of 33 cases) and a low Met (20/33). Elevated C3/C2 (7/33), high Met/Phe (5/33) and high C3 (4/33) were less indicative for a vB12D. Of the second-tier biomarkers, HCY was found to show the best sensitivity, being elevated in 30 of 33 cases, while MMA and MCA were only elevated in 12 and 9 patients, respectively [69]. Using post-analytical methods, such as multivariate pattern recognition software developed by Collaborative Laboratory Integrated Reports (CLIR—Mayo Clinic), also revealed a clear demarcation of the C3/Gly ratio from the reference range in pathologic conditions affecting Cbl metabolism [77,78]. Thus, current knowledge suggests that instead of a single marker, a combination of appropriately selected first-tier cutoffs of metabolites and ratios, together with second-tier confirmatory assays for the determination of HCY, MMA and MCA are expected to facilitate the diagnosis of a vB12D in NBS.

Cut-off values defined for the above biomarkers varied according to different algorithms used by NBS laboratories. A single marker (HCY high cut-off: 8 µmol/L) was only used in Norway [39]. Most laboratories used the classical parameters C3 (high cut-off: 3.3–5.5 µmol/L), C3/C2 (high cut-off: 0.18–0.26), and C3/C16 (high cut-off: 1.5–2.0) [39,69,79,80,81]. When second tier MMA and HCY measurements were not available, additional serum vB12, folic acid, HCY and urinary MMA determinations were performed [79]. Cut-off values for additional first tier markers were: C3/C0: >0.23; C3/Met: >0.2; Met: <8–11 µmol/L, Met/Phe < 0.19–0.56. MMA and HCY second tier high cuf-offs varied between 0.4–4 µmol/L and 10–12 µmol/L, respectively [69,79,80,81]. None of these protocols was compared in a single study on the same population to test their efficiency; however, all identified vB12D in newborns with a prevalence that was previously underestimated at national level.

## 9. Treatment and Follow-Up

Maternal vB12 supplementation during gravidity and post-partum (until the 6th week) could serve as primary prevention strategy not only through better Cbl values of the newborn at birth but also due to an increase of Cbl in breastmilk [82]. Mothers receiving oral vB12 supplementation gave birth to infants whose language skills were significantly better at 30 months of age than those in the un-supplemented group [58]. VB12D in and after the post-partum period might lead to irreversible neurologic changes from as early as 6 months of age [41,77]. Therefore, it is advisable to establish the diagnosis and start supplementation as early as possible.

Infant formulas are fortified by vB12 and serve as adequate tools for vitamin administration [1,44]. In the case of breastfed infants, vitamin supplementation of both the mother and the infant is recommended, if investigations reveal a maternal vB12D [44,83]. Before prescription, one should consider the type of Cbl available and the way it can be administered. The palette consisting of the traditional CNCbl was later extended with nature biosimilar OHCbl, AdoCbl, and MeCbl. The bioavailability of all forms is similar, though biosimilar products should be favored due to the lack of cyanide load for the body [84].

Infantile parenteral administration of Cbl might occur in the form of intramuscular injection (most common), or via intravenous injection (usually as part of the water-soluble vitamin complex required during parenteral feeding—common in pre-term infants). A single intramuscular injection (400 µg OHCbl in infants < 8 months) may correct the impaired Cbl status and improve the already evolving clinical symptoms, such as motor dysfunction and regurgitations [85]. The less invasive, longer lasting oral supplementation scheme could also be an alternative; starting with higher doses for 3 days (500 µg Cbl/day), followed by reduction to the fifth of the original dose until the complete recovery of vB12 levels and functional markers. A maintenance dose of 5 µg/day can be ceased after introduction of animal protein-containing complementary food. Oral administration scheme was complemented with folate substitution during the first week, and laboratory checkup at 2 weeks (possible exclusion of genetic defects in Cbl absorption) after supplementation onset in a recent study [44]. According to the personal opinion of the authors, socio-economic status, parenteral compliance, geographic distances needed for checkups should all be considered before choosing the way of vB12 administration. A genetic defect affecting vB12 absorption or metabolism should be considered if cessation of the treatment induces a relapse in laboratory markers [44,85].

## 10. Conclusions

The vB12D is a condition that can severely affect quality of life if not diagnosed early or left untreated. In low-income regions with high prevalence of vB12D and less financial resources for NBS, primary prevention, such as mandatory food fortification, or recommendations in parallel with educations for vitamin supplementation during pregnancy could be the best choice [86]. Recent studies suggest that an extension of the NBS (when already using MS/MS methods) with vB12D may be feasible, and even financially affordable. According to the opinion of the authors, a first trimester check-up of the vB12 status combined with NBS for vB12D could be an optimal way in reduction of the condition’s prevalence in women and infants of developed countries. Pregnancy screening alone is not sufficient as vitamin supplementation of pregnant women may be suboptimal compared to that of newborns due to the different effectivity of the social net in the two groups in most countries. For optimal outcomes, pregnancy screening should be complemented with NBS that is performed along standardized algorithms, equal for the entire population of a region or a country. VB12 deficient newborns identified by the NBS benefit remarkably from the early supplementation in asymptomatic state, resulting in the prevention of long-term consequences for the child, as well as additional health care benefits for the mother and reduced expenses and load for the health care system.

Standardized protocols and consensus guidelines for vB12D are yet to be developed and, together with recent improvements in laboratory technology and clinical treatment, are expected to facilitate earlier diagnosis and a better quality of life of patients with vB12D.

## Figures and Tables

**Figure 1 metabolites-12-01104-f001:**
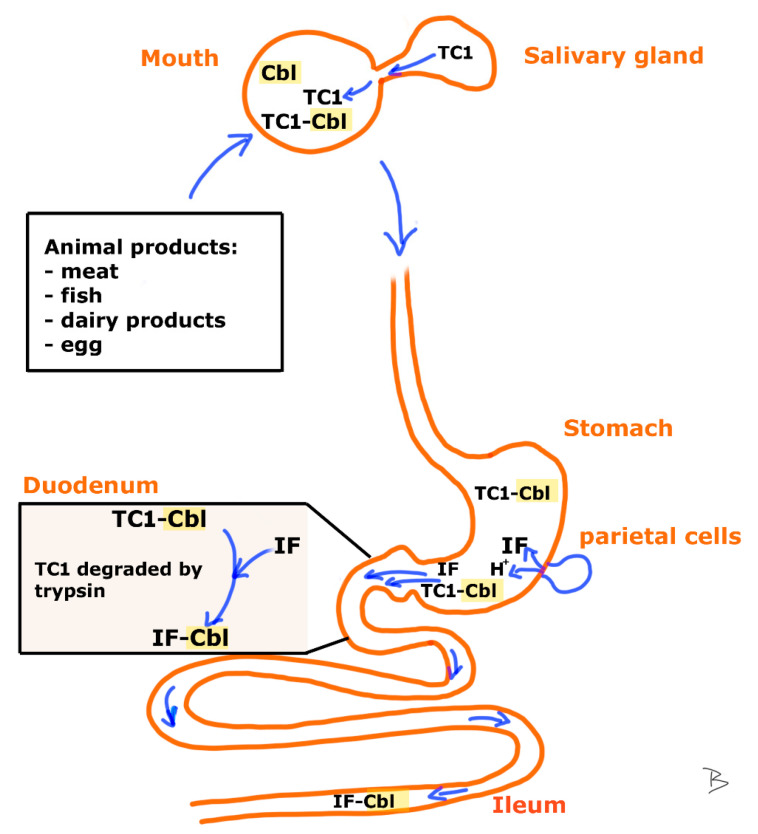
Route of Cbl from intake to the place of absorption. Dietary Cbl is anchored to the protein transcobalamin-1 (TC1, suggested terminology according to Uniprot search term: P20061 or TCO1_HUMAN, common used name: haptocorrin [8]) either in the mouth (TC1 of salivary gland origin) or already before ingestion, since human breastmilk contains TC1 bound Cbl (TC1-Cbl). Attachment to this protein prevents the damage of the cofactor at the acidic pH of the stomach. In the duodenal lumen pancreatic trypsin partially degrades TC1. Though secreted by the parietal cells of the stomach, intrinsic factor (IF) is only bound to Cbl in the duodenum, after it is released from TC1. Among factors affecting IF synthesis are gastrin, acetylcholine and histamine. The newly formed IF-Cbl complex is transported to the ileum where the apical membrane of the intestinal epithelial cells of the brush-border incorporates it via receptor mediated endocytosis.

**Figure 2 metabolites-12-01104-f002:**
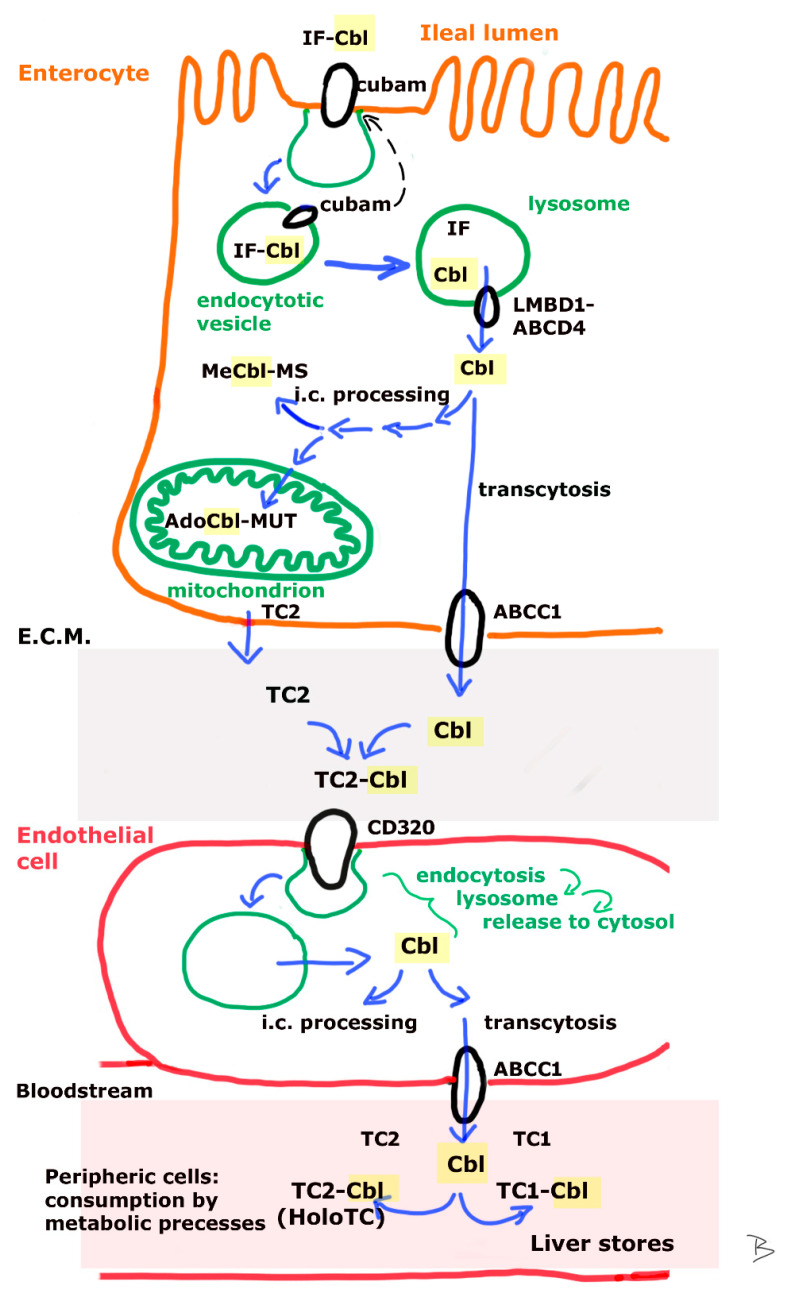
From absorption to distribution. The IF-Cbl complex is internalized after binding cubam (which is later recycled to the luminal membrane). In the endosome that is evolving to lysosome, the lowering of the pH makes IF dissociate from Cbl, which is then exported to the cytosol by the LMBD1-ABCD4 (lysosomal cobalamin transport escort protein complex—lysosomal cobalamin transporter ABCD4). Cytosolic free Cbl is either processed to active coenzymes for the enterocyte’s own metabolism or undergoes transcytosis, leaving the cell toward the extracellular matrix via ABCC1(ATP-binding cassette, subfamily C, member 1). Within the bloodstream it either takes the route of storage in the liver by binding TC1 (inactive vB12), or stays capable for uptake and utilization by the cells by binding TC2 (active vB12/holotrans-Cbl (HoloTC)).

**Figure 3 metabolites-12-01104-f003:**
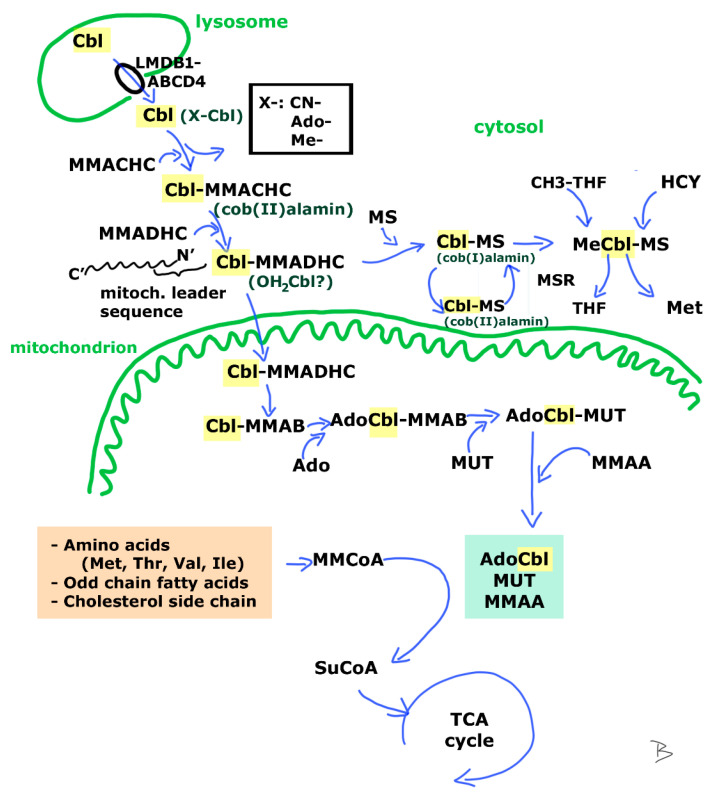
Intracellular Cbl processing. After leaving the lysosome via the LMDB1-ABCD4 complex, the Cbl in cytoplasm might be linked to various ligands (X-). During processing by MMACHC, ligand “X-“ is removed and the generated cob(II)alamin is further oxidized by MMADHC. This latter contains an N’-terminal mitochondrial leader sequence and is capable of directing the attached Cbl either to the cytoplasm, or to the mitochondria. In the cytoplasm the complex donates Cbl to MS that becomes capable for receiving the Me-group of CH3-THF and converting HCY to Met. The active form of MeCbl-MS contains cob(I)alamin that might spontaneously oxidize to cob(II)alamin leading to inactive MS. Cob(II)alamin is regenerated by the enzyme Met synthase reductase (MSR). In the mitochondria Cbl in the Cbl-MMADHC complex is processed by MMAB enabling the adenosylation of Cbl. The so-formed AdoCbl connects to MUT. Together with MMAA the complex is capable of the enzymatic transition of MMCoA to succinyl CoA (SuCoA) [12,13]. For explanation of protein symbols see Table 1 below. CH3-THF methyl tetrahydrofolate reductase; OH_2_Cbl aquo-Cbl; TCA tricarboxylic acid.

**Figure 4 metabolites-12-01104-f004:**
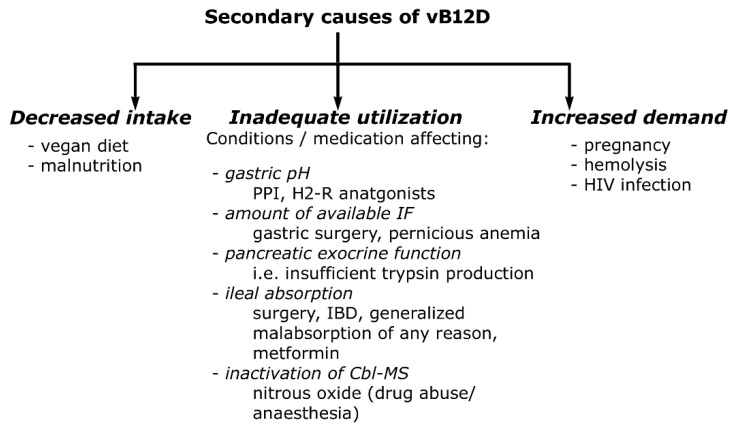
Secondary causes of vB12 deficiency. PPI proton pump inhibitor; H2-R histamine2-receptor; IBD inflammatory bowel disease; HIV human immunodeficiency virus.

**Table 1 metabolites-12-01104-t001:** Inherited Cbl deficiencies. For current protein nomenclature we refer to UniProt database [8]. Letters in the gene symbol referring to the certain letter in the disease name are marked with bold in case of Cbl defects: CblA, CblB, CblC, and CblD. HCYuria homocystinuria; I.c. intracellular; Lysos. membr. lysosomal membrane; MMAuria methylmalonic aciduria; Mitoch. mitochondria.

I.c. Locus	Protein Name		Gene Symbol	Clinical Phenotype
	Alternative (Common)	Recommended		
Lysos. membr.	Lysosomal transport escort protein LMBD1	LMBD1	LMBDR1	CblF defect
	Lysosomal Cbl transporter ABCD4	ABCD4	ABCD4	CblJ defect
Cytosol	CblC, MMACHC, MMAuria and HCYuria type C protein	CNCbl reductase/AlkylCbl dealkylase	MMA**C**HC	Cbl**C** defect
Cytosol/Mitoch.	CblD, MMADHC, MMAuria and HCYuria type D protein	Cobalamin trafficking protein CblD	MMA**D**HC	Cbl**D** defect
Mitochondria	Methylmalonic aciduria type B protein	Corrinoid adenosyltransferase MMAB	MMA**B**	Cbl**B** defect
	MMAA (Methylmalonic aciduria type A protein, mitochondrial)	Methylmalonic aciduria type A protein, mitochondrial	MMA**A**	Cbl**A** defect

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
