# Peer review of "Vitamin B12 (Cobalamin): Its Fate from Ingestion to Metabolism with Particular Emphasis on Diagnostic Approaches of Acquired Neonatal/Infantile Deficiency Detected by Newborn Screening"

_metabolites, 2022, doi:10.3390/metabo12111104_

Round 1
Reviewer 1 Report
This review article by Kosa et al. gives an overview over the important and interesting topic of vitamin B12 deficiency with a focus on infants, particularly newborns. It is well written and combines description of basic mechanisms of cobalamin action with current developments and controversies on detection of vitamin B12 deficiency via newborn screening.
Just two important points that should be improved:
Chapter 5: Diagnostic criteria for vitamin B12 deficiency should be discussed more thoroughly in the light of (the difficult) definition of vitamin B12 deficiency in newborns. In other words: how should vitamin B12 deficiency be defined in newborns including reference ranges or cut-offs that should be applied according to the available literature as well as the expertize of the authors.
Chapter 8: Since there is published data from Spain (2 different regions), Estonia, Austria, Norway, Germany (as cited) and maybe others, the different screening algorithms should be discussed in respect to the big differences of their outcomes. I would like to encourage the authors to be more concrete on how a newborn screening for vitamin B12 should be performed.
Minor:
It should not be left unmentioned, that the optimal preventive measures were information, screening and supplementation of pregnant women.
Author Response
Thank you for your valuable work, and important suggestions, on how the manuscript should be further improved and extended. We also thank you for encouraging us to share our personal opinion and expertise.
Chapter 5: This chapter is basically dedicated to discussing diagnostic possibilities and difficulties of vB12D. The approaches discussed here could be used during the identification of maternal vB12D, or in cases of NBS programs where second-tier tests (HCY, MMA from DBS) are not possible. As requested, different considerations including vB12 cut-offs were included in the main text under “5.5. General considerations” (page 9), with our stressed remark, that those cut-offs should be used as a guide only and cannot be directly used in other settings, due to the variety of assay reference intervals used in different laboratories. This is in accordance with our personal experience in Hungary when working with the results of several clinical laboratories, using different kits and reference intervals. The different approaches to diagnostic criteria were included in Chapter 8.
Chapter 8: as requested, exact cut-offs used by different screening programs (Austria, Germany, Italy, Estonia) were included according to the suggestion of the reviewer in the last paragraph of the chapter (page 12)
Minor: As it was requested, the importance of education, screening, and supplementation of pregnant women was emphasized in the manuscript (Chapter 10. Conclusions, page 13).
Reviewer 2 Report
Thank you very much for letting me participate in the review of this interesting article. The precision and well-documented nature of the first sections stands out, particularly sections 2, 3 and 4. One could try to reduce the length of the same sections by exclusively selecting aspects that are relevant for the subsequent discussion regarding diagnosis and treatment if the length of the work was a problem.
It seems to me that the presentation of the data presented opens the discussion, but as the authors very well recognize, more background is required to decide to include this condition in newborn screening (NBS) programs at the national level in different countries. It seems necessary for this discussion to expand on the technical limitations, as well as from the perspective of the most appropriate selection of conditions for NBS programs. As an example of the latter, it would be interesting to know the opinion of the authors, knowing that in section 6 they state that this condition occurs "especially in low-income areas where access to animal food is strongly limited", if it would be cost-benefit to start the NBS with this condition, before other conditions such as congenital hypothyroidism, phenylketonuria or medium-chain fatty acid oxidation defect (MCAD), which have demonstrated their cost-effectiveness for many years, in countries where the selection of conditions must be very careful, understanding that resources are more limited than in countries that already carry out screening for more than 20 or 30 conditions.
Another interesting discussion is what the authors refer to in section 9, regarding the fact that the greatest impact of the treatment would be achieved "during pregnancy and post-partum", so that the NBS result could arrive late at the optimal moment for its treatment.
These aspects, together with others related to cost effectiveness, seem to me relevant for decision makers in different countries and socioeconomic conditions that allow them to decide to choose this or another condition to be incorporated into NBS programs.
A third aspect to discuss are the ethical aspects related to the autonomy of mothers who use restrictive diets, mentioned as an important cause of this condition. One could imagine that educating women of childbearing age regarding these risky behaviors could have some impact in preventing the condition discussed in this article.
Author Response
Thank you for your valuable work, consideration, and encouraging opinion about our work.
The first few chapters were not edited since there was no editorial request in this direction. The aim of the manuscript was to provide a global overview of vB12D for colleagues in NBS who may also meet inherited cobalamin-related disorders. If so, the details of these sections could be of importance.
The opinion of the authors considering optimal strategies for developing countries with an extremely high prevalence of vB12 compared to other developed areas was included in the “Conclusions” section (page 13). Unfortunately, comprehensive and accurate information about the financial costs of screening (during pregnancy or in infants), food fortification, or health expenses in different parts of the globe is lacking. Thus, even if it seems to be highly probable that the prevention of vB12D is cost-effective, this cannot currently be stated.
We reflected on the possibility that the Reviewer raised: “ the NBS result could arrive late at the optimal moment for its treatment.” in the “Conclusions” section (page 13).
The question of the mothers’ autonomy was in part added to the “Conlcusions” section. (page 13).
Reviewer 3 Report
Vitamin B12 deficiency (vB12D) is one of the disorders which, if not diagnosed in time, fundamentally affects early development, health and quality of life. It thus fulfills the basic condition for the introduction of its screening. In addition to relatively rare genetically determined vB12D disorders, it is caused by insufficient intake of B12 during pregnancy and breastfeeding - as a result of social state, but also alternative dietary habit. The authors present a comprehensive view of the issue of vB12D - from an overview of the metabolic pathways of vB12, possible markers for detection and neonatal screening (NS), they also discuss the problem of detecting a deficiency during pregnancy and during breastfeeding. NS vB12D is shown to be up-to-date using already established NBS methodologies (ELISA, ILMA, MS/MS) for other inborn errors of metabolism (IEM). The specificity of vB12D from the NBS point of view is, in addition to the hereditary disorder, its exogenous, socially conditioned origin with the possible need for comprehensive preventive vB12 supplementation during pregnancy and during breastfeeding, analogous to vB9 supplementation in the prevention of neural tube defect (NTD). The manuscript meets the conditions for its publication.
Author Response
We were pleased to read your opinion about our manuscript. Thank you very much for your valuable work, and inspiring summary.